# Serendipity and strategy in rapid innovation

T.M.A. Fink[1,2], M. Reeves[3], R. Palma[3] & R.S. Farr[1]

Innovation is to organizations what evolution is to organisms: it is how organizations adapt to environmental change and improve. Yet despite advances in our understanding of evolution, what drives innovation remains elusive. On the one hand, organizations invest heavily in systematic strategies to accelerate innovation. On the other, historical analysis and individual experience suggest that serendipity plays a significant role. To unify these perspectives, we analysed the mathematics of innovation as a search for designs across a universe of component building blocks. We tested our insights using data from language, gastronomy and technology. By measuring the number of makeable designs as we acquire components, we observed that the relative usefulness of different components can cross over time. When these crossovers are unanticipated, they appear to be the result of serendipity. But when we can predict crossovers in advance, they offer opportunities to strategically increase the growth of the product space.

[1] London Institute for Mathematical Sciences, 35a South St, Mayfair, London W1K 2XF, UK. [2] Centre National de la Recherche Scientifique, Paris, 75005, France. [3] The Boston Consulting Group, BCG Henderson Institute, New York, 10016 NY, USA. Correspondence and requests for materials should be addressed to T.M.A.F. (email: tmafink@gmail.com)

Innovation is how governments, institutions and firms adapt to changes in the environment and improve[1]. Organizations that innovate are more likely to prosper and stand the test of time; those that fail to do so fall behind their competitors and succumb to market and environmental change[2, 3]. Despite the importance of innovation, what drives innovation remains elusive[1, 4]. Research on macro-economic development suggests that more complex, diverse or re-purposable[5–8] production capabilities result in greater economic growth at a national level[9, 10]. At a micro-economic level, there is a perennial tension between a strategic approach, which views innovation as a rational process which can be measured and prescribed[11–13]; and a belief in serendipity and the intuition of extraordinary individuals[14–16].

The strategic approach is seen in firms like P&G and Unilever, which use process manuals and consumer research to maintain a reliable innovation factory[17], and Zara, which systematically scales new products up and down based on real-time sales data. In scientific discovery, "traditional scientific training and thinking favour logic and predictability over chance"[14]. If discoveries are actually made in the way that published papers suggest, the path to invention is a step-wise, rational process.

On the other hand, a serendipitous approach is seen in firms like Apple, which is notoriously opposed to making innovation choices based on incremental consumer demands, and Tesla, which has invested for years in their vision of long-distance electric cars[18]. In science, many of the most important discoveries have serendipitous origins, in contrast to their published step-by-step write-ups, such as penicillin, heparin, X-rays and nitrous oxide[14]. The role of vision and intuition tend to be under-reported: a study of 33 major discoveries in biochemistry "in which serendipity played a crucial role" concluded that "when it comes to 'chance' factors, few scientists 'tell it like it was'"[19, 20].

To unify these two perspectives and understand what drives innovation, in this Article we do four things. First, we study data from three sectors: language, gastronomy and technology. We measure how the number of makeable products (words, recipes and software products) grows as we acquire new components (letters, ingredients and development tools). We observe that the relative usefulness of components is not fixed, but cross each other in time. Second, to explain these crossovers, we prove a conservation law for the innovation process over time. The conserved quantity is a combination of the usefulness of components and the complexity of products. We use it to forecast crossovers in the future based on information we already have about the products we can make. Third, we identify a spectrum of innovation strategies dependent on how far into the future we forecast: from short-term gain to long-term growth. A short-sighted strategy maximizes what a new component can do for us now, whereas a far-sighted strategy maximizes what it could do for us later. We apply both strategies to our three sectors and find that they differ from each other to the extent that each sector contains crossovers. Fourth, we resolve the tension between the strategic and serendipitous interpretations of innovation. Both can be viewed as different manifestations of the changing importance of components over time. When component crossovers are unexpected, they appear to stem from serendipity. But when we can forecast crossovers in advance, they provide an opportunity to strategically increase the growth of our product space.

## Results

**Lego game**. We begin by illustrating our ideas using Lego bricks. Think back to your childhood days. You're in a room with two friends Bob and Alice, playing with a big box of Lego bricks—say, a fire station set. All three of you have the same goal: to build as many new toys as possible. As you continue to play, each of you searches through the box and chooses those bricks that you believe will help you reach this goal. Let us now suppose each player approaches this differently. Your approach is to follow your gut, arbitrarily selecting bricks that look intriguing. Alice uses what we call a short-sighted strategy, picking Lego men and their firefighting hats to immediately make simple toys.

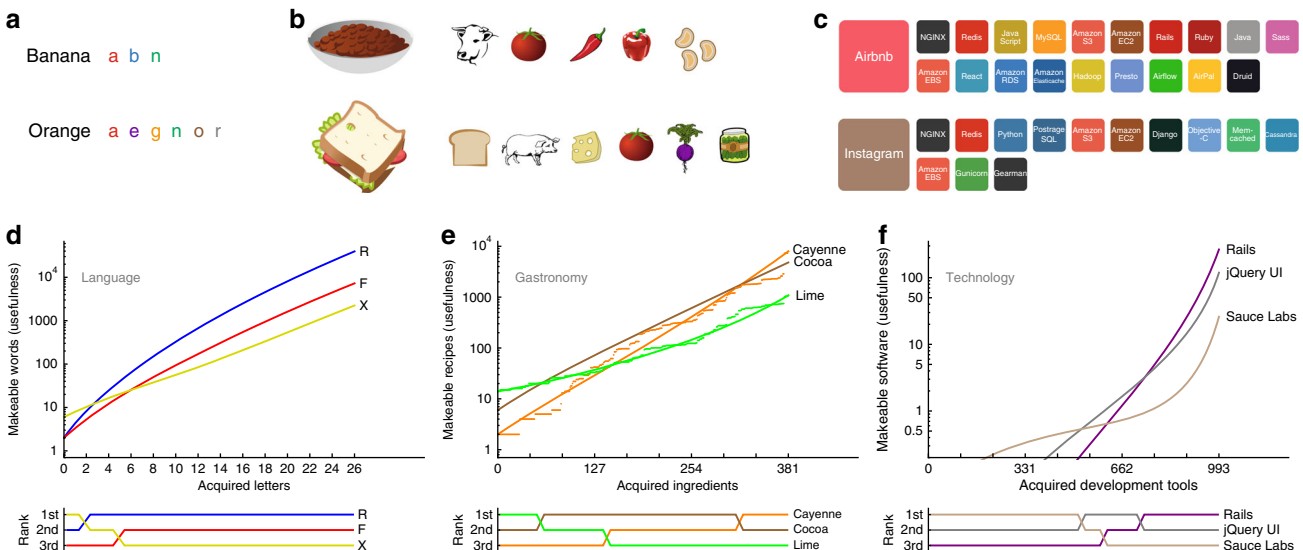

**Fig. 1** Products, components and usefulness for three sectors. We studied products and components from three sectors. **a** In language, the products are 79,258 English words and the components are the 26 letters. **b** In gastronomy, the products are 56,498 recipes from the databases allrecipes.com, epicurious.com and menupan.com[23], and the components are 381 ingredients. **c** In technology, the products are 1158 software products catalogued by stackshare.io and the components are 993 development tools used to make them. **d–f** The usefulness of a component is the number of products we can make that contain it. We find that the relative usefulness of a component depends on how many other components have already been acquired. For each sector, we show the usefulness of three typical components: averaged at each stage over all possible choices of the other acquired components and—for gastronomy—for a particular random order of component acquisition (points)

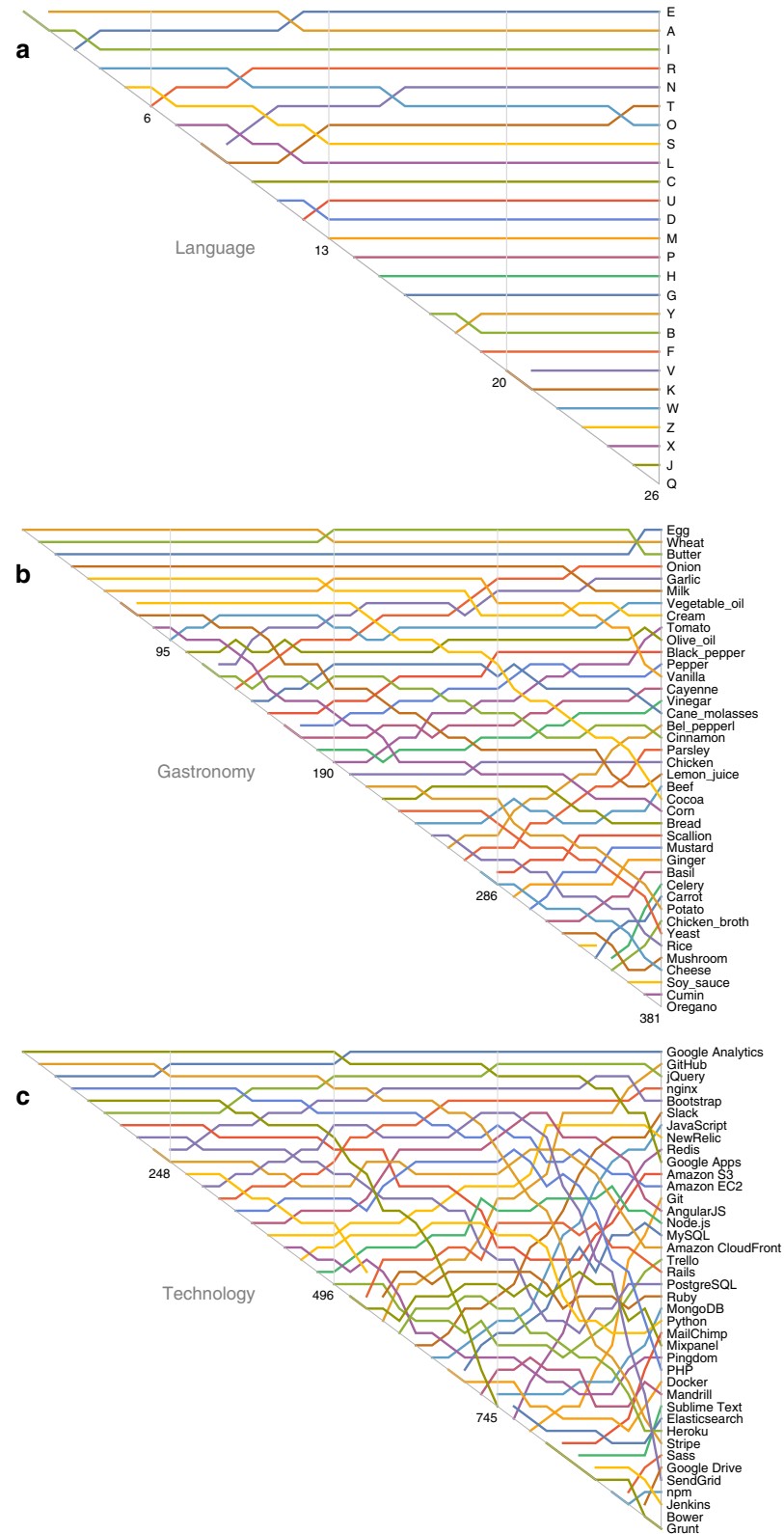

**Fig. 2 a–c** Crossovers in the mean usefulness of components. The relative usefulness of different components changes as the number of components we possess increases. For example **a**, if you are only allowed six letters, the ones that show up in the most words are *a, e, i, o, s, r*. For gastronomy **b** and technology **c**, for clarity we only show the 40 components most useful when we have all *N* components. A pure short-sighted strategy attempts to acquire components in the order that they intersect the diagonal, whereas a pure far-sighted strategy attempts to acquire them in the order that they intersect a vertical. If there are no crossovers, the strategies are the same

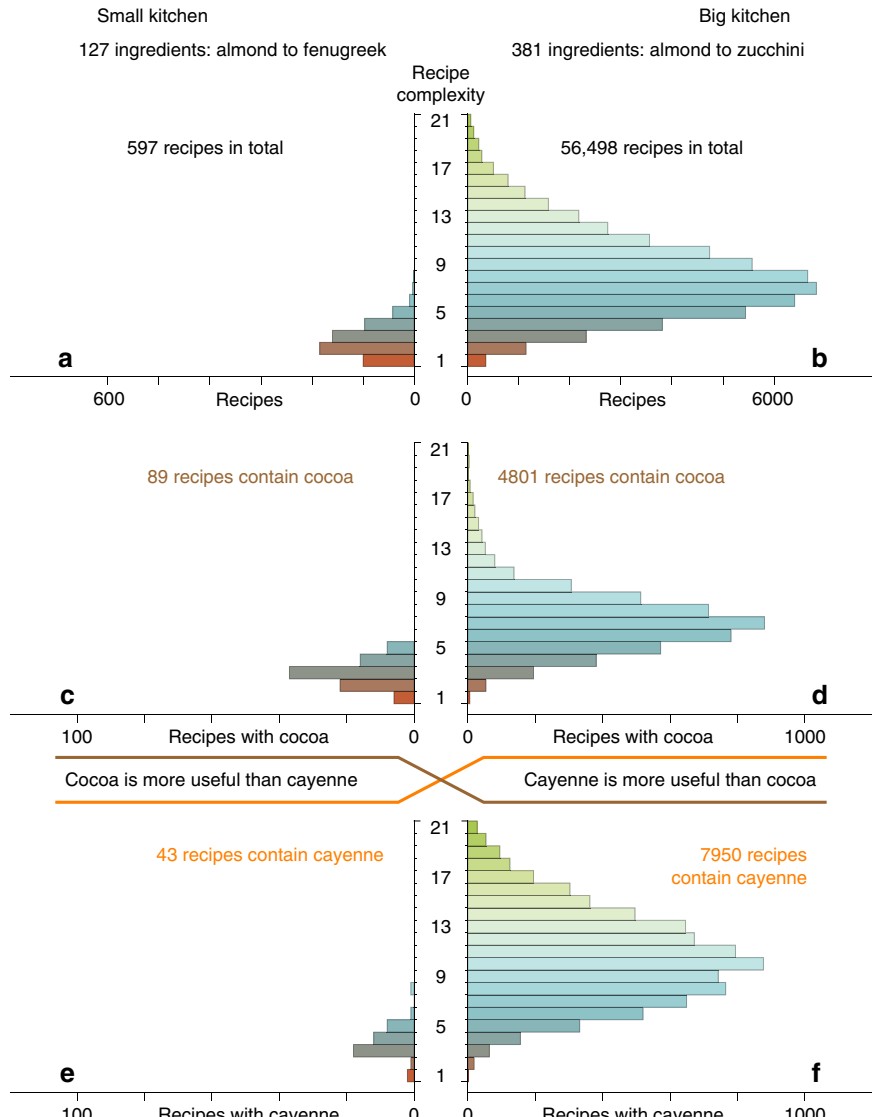

**Fig. 3** Recipes broken down by complexity, showing why crossovers happen. On the right is a big kitchen with 381 ingredients. On the left is a small kitchen with one-third as many ingredients. In the big kitchen **b**, we can make a total of 56,498 recipes. Each bar counts recipes with the same number of ingredients (complexity). When we move to the smaller kitchen **a**, the number of makeable recipes shrinks dramatically to 597, or 1.0%. But this reduction is far from uniform across different bars. Higher bars shrink more, on average by an extra factor of 3 with each bar. Thus, the number of recipes of complexity one (first bar) shrinks about threefold; the number of complexity two (second bar) ninefold, and so on. Of all the recipes in the big kitchen, 4801 contain cocoa **d** and 7950 contain cayenne **f**. The cayenne recipes tend to be more complex, containing on average 10.6 ingredients, whereas the cocoa recipes are simpler, averaging 7.2 ingredients. Because higher bars suffer stronger reduction, overall fewer cayenne recipes (0.5%) survive in the smaller kitchen **e** than cocoa recipes (1.8%) **c**. Thus, cayenne is more useful in the big kitchen, but cocoa is more useful in the small kitchen

Meanwhile, Bob chooses pieces such as axles, wheels and small base plates that he noticed are common in more complex models, even though he is not able to use them straightaway to produce new toys. We call this a far-sighted strategy.

**Who wins**. At the end of the day, consider who will have innovated the most, by building the most new toys. We find that, in the beginning, Alice will lead the way, surging ahead with her impatient strategy. But as the game progresses, fate will appear to shift. Bob's early moves will begin to look serendipitous when he is able to assemble a complex fire truck from his choice of initially useless axles and wheels. It will seem that he was lucky, but we will soon see that he effectively created his own serendipity. As for you, picking components on a hunch, you will have built the

fewest toys. Your friends had an information-enabled strategy, while you relied on chance.

**Spectrum of strategies**. The Lego example highlights an important concept. If innovation is a search process, then your component choices today matter greatly in terms of the options they will open up to you tomorrow. You can pick components that quickly form simple products and give you a return now, or you can choose those components that give you a higher future option value. By understanding innovation as a search for designs across a universe of components, we made a surprising discovery. Information about the unfolding process of innovation can be used to form an advantageous innovation strategy. But there is no one superior strategy. As we shall see, the optimal strategy

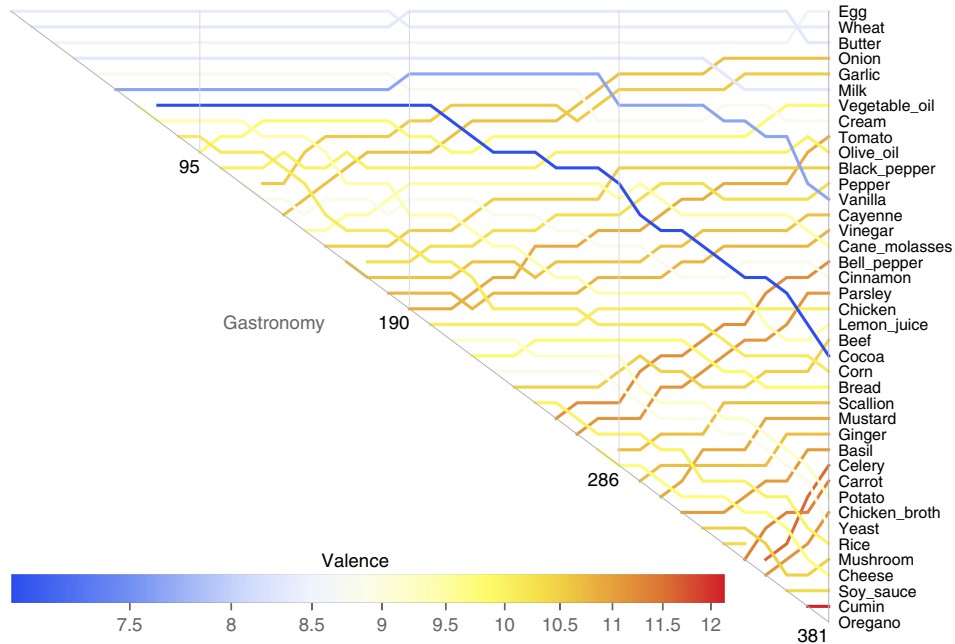

**Fig. 4** How valence affects the relative usefulness of components over time. Here we show Fig. 2b, but with the component curve colours set by component valence at stage N. More valent components tend to rise in relative usefulness, and less valent components tend to fall

depends on time—how far along the innovation process we have advanced—and the sector—some sectors contain more opportunities for strategic advantage than others.

**Components and products**. Just like the Lego toys are made up of distinct kinds of bricks, we take products to be made up of distinct components. A component can be an object, like a touch screen, but it can also be a skill, like using Python, or a routine, like customer registration. Only certain combinations of components form products, according to some predetermined universal recipe book of products. Examples of products and the components used to make them are shown in Fig. 1. Now suppose that we possess a basket of distinct components, which we can combine in different ways to make products. We have more than enough copies of each component for our needs, so we do not have to worry about running out. There are $N$ possible component types in total, but at any given stage $n$ we only have $n$ of these $N$ possible building blocks. At every stage, we pick a new type of component to add to our basket.

**Usefulness**. The usefulness of a component is the number of products we can make that contain it. In other words, the usefulness $u_\alpha$ of some component $\alpha$ is how many more products we can make with $\alpha$ in our basket than without $\alpha$ in our basket. As we gather more components, $u_\alpha$ increases or stays the same; it cannot decrease. We write $u_\alpha(n)$ to indicate this dependence on $n$: $u_\alpha(\boldsymbol{n})$ is the usefulness of $\alpha$ given possession of $\alpha$ and a specific set of $n-1$ other components, the combined set of components being $\boldsymbol{n}$. Averaging over all choices of the $n-1$ other components from the $N-1$ that are possible gives the mean usefulness, $\bar{u}_\alpha(n)$. We make no assumptions about the values of different products, which will depend on the market environment and may change with time. But we can be sure that maximizing the number of products is a proxy for maximizing any reasonable property of them. A similar proxy is used in evolutionary models, where evolvability is defined as the number of new phenotypes in

the adjacent possible (1-mutation boundary) of a given phenotype; see ref. [21].

**Usefulness experiment**. To measure the mean usefulness of different components as the innovation process unfolds and we acquire more components, we did the following experiments. Using data from each of our three sectors, we put a given component $\alpha$ into an empty basket, and then added, one component at a time, the remaining $N-1$ other components, measuring the usefulness of $\alpha$ at every step. We averaged $u_\alpha(\boldsymbol{n})$ over all possible orders in which to add the $N-1$ components to obtain $\bar{u}_\alpha(n)$. (We explain how in "Proof of components invariant" in the Methods section.) We repeated this process for all of the components $\alpha$. Results for typical components are shown in Fig. 1. We find that the mean usefulnesses of different components cross each other as the number of components in our basket increases. As Fig. 1 shows for gastronomy, this is true for both the mean usefulness and the usefulness itself, measured for a specific random ordering of components (points). What the mean usefulness does, the usefulness tends to do also, because it is an unbiased estimate of its mean.

**Bumps charts**. To visualize the relative usefulness of components over time, for each sector we created its "bumps chart" (Fig. 2 and, more complete, in Supplementary Figs. 1b and 2b). These show the rank order of mean usefulness at every stage of the innovation process. We see that the crossovers in Fig. 1 are commonplace, but that some sectors contain more crossovers than others. There are few crossings in language, some in gastronomy and many in technology. This means, for example, that the most useful letters for making words in Scrabble (a basket of seven letters) are nearly the same as the most useful letters for making words with a full basket (26 letters); the key ingredients in a small kitchen (20 ingredients) are moderately different from those in a big one (80 ingredients); the most-used development skills for a young software firm (experience with 40 tools) are significantly different from those for an advanced one (160 tools).

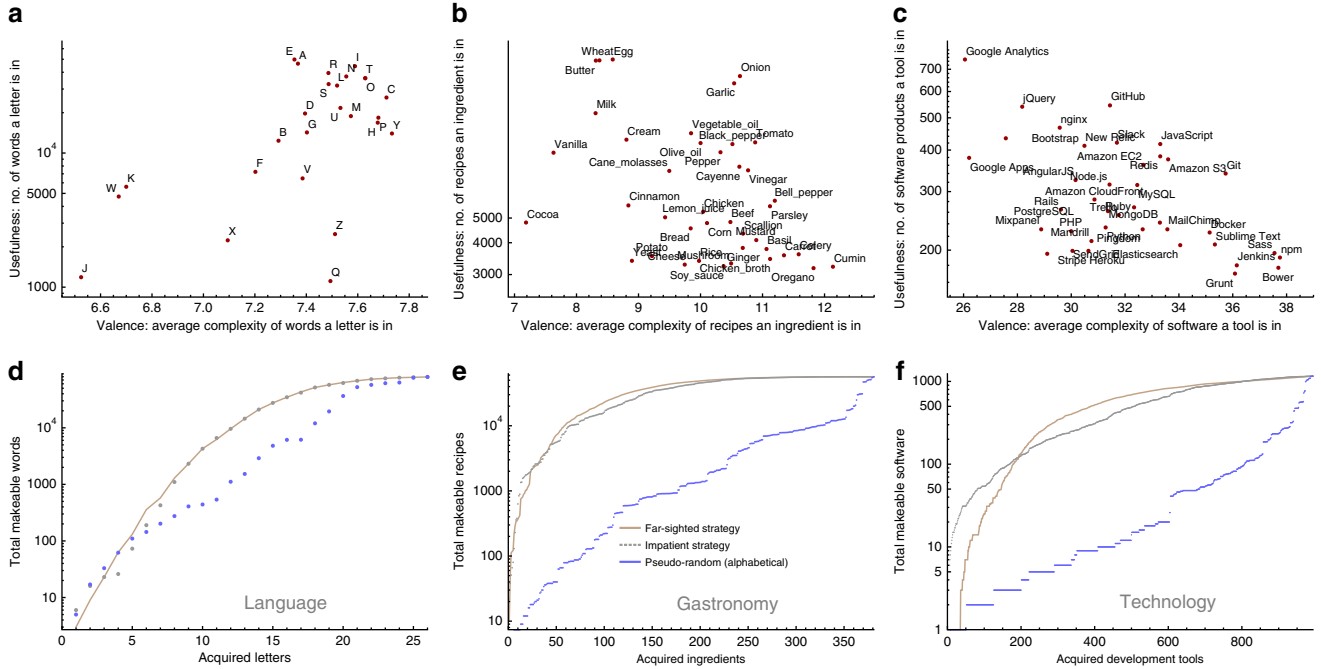

**Fig. 5** The success of different innovation strategies depends on the distribution of component properties. **a–c** Scatter plots of component mean usefulness vs. component valence for our three sectors. For gastronomy and technology, we only show the top 40 components. **d–f** Both the short-sighted and far-sighted strategies beat a typical random component ordering (here alphabetical), but they diverge from each other only insofar that there are crossings in the bumps charts

**Conservation law**. To understand why crossovers happen, let us have a closer look at how the usefulness increases for a single component (Fig. 3). To make a product of complexity $c$, we must possess all $c$ of its distinct components. So making a complex product is harder than making a simple one, because there are more ways that we might be missing a necessary component. We therefore group together the products we can make containing $\alpha$ according to their complexity. That is, the usefulness $u_\alpha(n, c)$ of component $\alpha$ is how many more products of complexity $c$ we can make with $\alpha$ in our basket than without $\alpha$ in our basket. Summing $u_\alpha(n, c)$ over $c$ gives $u_\alpha(n)$. The advantage of this refined grouping is that, by understanding the behaviour of $\overline{u}_\alpha(n, c)$, we can understand the more difficult $\overline{u}_\alpha(n)$. Our key result, which we prove in the Methods section, is that $\overline{u}_\alpha(n, c)\, n / \binom{n}{c}$ is constant over all stages of the innovation process, where $\binom{n}{c}$ is the binomial coefficient. In other words, for two stages $n$ and $n'$,

$$\overline{u}_\alpha(n, c)\, n / \binom{n}{c} = \overline{u}_\alpha(n', c)\, n' / \binom{n'}{c}. \quad (1)$$

Solving this for $\overline{u}_\alpha(n', c)$, this tells us that the number of products containing $\alpha$ of complexity $c$ grows much faster for higher complexities than for lower complexities. Early on, $\overline{u}_\alpha(n, c)$ will tend to be small for higher complexities, but depending on how far ahead we look, the bigger growth rate can more than compensate for this, as we see in Fig. 3. Summing Eq. (1) over size $c$ and approximating $\binom{n}{c}$ and $\binom{n'}{c}$ by $n^c$ and $n'^c$, we find

$$\overline{u}_\alpha(n') \simeq \overline{u}_\alpha(n, 1) + \overline{u}_\alpha(n, 2)\, x + \overline{u}_\alpha(n, 3)\, x^2 + \ldots, \quad (2)$$

where $x = n'/n$. Because the usefulness is an unbiased estimate of its mean, we can approximate $\overline{u}_\alpha(n, c)$ on the right-hand side of

Eq. (2) with $u_\alpha(n, c)$, which we know at the present time. With these substitutions, we use Eq. (2) to calculate $\overline{u}_\alpha(n')$ at some point in the future. This is then an estimator for the usefulness for any superset of $\boldsymbol{n}$ of size $n'$ (a superset because we can only add components to our basket). This enables us to make predictions about future usefulness entirely from information we have in the present.

**Valence**. So far we have only characterized a component by its usefulness: the number of products we can make that contain it. Now we introduce another way of describing a component: the average complexity of the products it appears in. We call this the valence, and it affects (Fig. 4) how the importance of components changes over time. The valence $v_\alpha(\boldsymbol{n})$ of component $\alpha$ is the average complexity of the products it appears in at stage $n$. Think of the valence as the typical number of co-stars a component performs with, plus one. We show the usefulness and valence at stage $N$ for different components in Fig. 5a–c and, more complete, in Supplementary Figs. 1a and 2a. More valent components are unlikely to be useful until we possess a lot of other components, so that we have a good chance of hitting upon the ones they need. These are the wheels and axles in our Lego set. On the other hand, less valent components are likely to boost our product space early on, when we have acquired fewer components. These are the Lego men and their firefighting hats. This insight suggests that more valent components will tend to rise in relative usefulness, and less valent components fall. This is verified in our experiments: components on the right of the plots in Fig. 5a–c tend to rise in the bumps charts, such as onion, tomato, Javascript and Git, whereas components on the left tend to fall, like cocoa, vanilla, Google Apps and SendGrid. Figure 4 shows this effect visually.

## Discussion

A crossover in the usefulness of components means that the things that matter most today are not the same as the things that will matter most tomorrow. How we interpret crossovers in practice depends on whether they are unanticipated, and take us by surprise, or anticipated, and can be planned for and exploited. When they are unanticipated, beneficial crossovers can seem to be serendipitous. But when they can be anticipated, crossovers provide an opportunity to strategically increase the growth of our product space. To harness this opportunity, we turn to forecasting component crossovers using the complexity of products containing them.

To maximize the size of our product space when crossovers are unanticipated, the optimal approach is to acquire, at each stage, the component that is most useful now. Think of this as a "greedy", or short-sighted, approach. It has a geometric interpretation: it is attempting to acquire the components that intersect the diagonals in Fig. 2. At every stage we lock in to a specific component, unaware of the future implications of the choices we make. A component poorly picked is an opportunity lost.

Using only information about the products we can already make with our existing components, however, we can forecast the usefulness of our components into the future applying a far-sighted strategy. Equation (2) shows us how, and we give an example in "Forecasting crossovers in usefulness" in the Methods section. Here the optimal approach is to acquire, at each stage, the component that will be most useful at some later stage $n'$. This also has a geometric interpretation: it is attempting to acquire the components that intersect a vertical at $n'$ in Fig. 2, and thus depends on how far into the future we forecast.

A short-sighted strategy considers only the usefulness $u_\alpha$, whereas a far-sighted strategy considers both the usefulness $u_\alpha$ and the valence $v_\alpha$. Short-sighted maximizes what a potential new component can do for us now, whereas far-sighted maximizes what it could do for us later. Depending on our desire for short-term gain vs. long-term growth, we have a spectrum of strategies dependent on $n'$. A pure short-sighted strategy ($n' = n$) and a pure far-sighted strategy ($n' = N$) are compared in Fig. 5d–f. Like the Lego approaches of Bob and Alice, both strategies beat acquiring components at random. As our theory predicts, the extent to which the two strategies differ from each other increases with the number of crossovers. For language, they are nearly identical, because there are few crossovers. For gastronomy, short-sighted has a twofold advantage at first, but later far-sighted wins by a factor of two. For technology, short-sighted surges ahead by an order of magnitude, but later far-sighted is dominant. While we do not fully understand why some datasets have more crossovers than others, our results suggest that a dataset with a broader spectrum of valences tends to have more crossovers. For language, gastronomy and technology the standard deviations of valence $v_\alpha(\mathbf{N})$ are 0.32, 2.0 and 18.

Writing about the *The Three Princes of Serendip*, Horace Walpole records that the princes "were always making discoveries, by accidents and sagacity, of things they were not in quest of". Serendipity is the fortunate development of events, and many organizations and researchers stress its importance[14, 15]. Crossovers in component usefulness help us see why. Components which depend on the presence of many others can be of little benefit early on. But as the innovation process unfolds and the acquired components pay off, the results will seem serendipitous, because a number of previously low-value components become invaluable. Thus, what appears as serendipity is not happenstance but the delayed fruition of components reliant on the presence of others. After the acquisition of enough other components, these components flourish. For example, the initially useless axles and wheels were later found to be invaluable to

building many new toys. In a similar way, the low value attributed to Flemming's initial identification of lysozyme was later revised to high value in the years leading to the discovery of penicillin, when other needed components emerged, such as sulfa drugs which showed that safe antibiotics are possible[14]. Interestingly, the word "serendipity" does not have an antonym. But as our bumps charts show, for every beneficial shift in a crossover, there is a detrimental one. Each opportunity for serendipity goes hand-in-hand with a chance for anti-serendipity: the acquisition of components useful now but less useful later. Avoiding these over-valued components is as important as acquiring under-valued ones to securing a large future product space.

Our research shows that the most important components— materials, skills and routines—when an organization is less developed tend to be different from when it is more developed. The relative usefulness of components can change over time, in a statistically repeatable way. Recognizing how an organization's priorities depend on its maturity enable it to balance short-term gain with long-term growth. For example, our insights provide a framework for understanding the poverty trap. When a less-developed country imitates a more-developed country by acquiring similar production capabilities[7], it is unable to quickly reap the rewards of its investment, because it does not have in place enough other needed capabilities. This in turn prevents it from further investment in those needed components. Our analysis gives quantitative backing to the "lean start-up" approach to building companies and launching products[22]. Start-ups are wise to employ a short-sighted strategy and release a minimum viable product. Without the resources to sustain a far-sighted approach, they need to quickly bring a simple product to market. On the other hand, firms that can weather an initial drought will see their sacrifice more than paid off when their far-sighted approach kicks in. By tracking how potential new components combine with existing ones, organizations can develop an information-advantaged strategy to adopt the right components at the right time. In this way they can create their own serendipity, rather than relying on intuition and chance.

## Methods

**Data**. Our three data sets—described in Fig. 1—were obtained as follows. In language, our list of 79,258 common English words is from the built-in WordList library in Mathematica 10.4. Of the 84,923 KnownWords, we only considered those made from the 26 letters a–z, ignoring case: we excluded words containing a hyphen, space, etc. In gastronomy, the 56,498 recipes can be found in the Supplementary Material in ref. [23]. In technology, the 1158 software products and the development tools used to make them can be found at the site stackshare.io.

**Proof of components invariant**. Let $\mathbf{N}$ be the set of $N$ possible components, let $\alpha$ be one of those components, and let $\mathbf{N}_1$ be the set of $N - 1$ other components not including $\alpha$. Let $\mathbf{n}_1$ be a subset of $n - 1$ components chosen from $\mathbf{N}_1$, and let $\mathbf{c}_1$ be a subset of $c - 1$ components chosen from $\mathbf{n}_1$. The usefulness $u_\alpha(\mathbf{n}, c)$ is how many more products of complexity $c$ that we can make from the components $\mathbf{n}_1$ together with $\alpha$, than from the components $\mathbf{n}_1$ alone:

$$u_\alpha(\mathbf{n}, c) = \sum_{\mathbf{c}_1 \subseteq \mathbf{n}_1} \mathrm{prod}(\alpha \cap \mathbf{c}_1) - \mathrm{prod}(\mathbf{c}_1), \tag{3}$$

where $\mathrm{prod}(\alpha \cap \mathbf{c}_1)$ takes the value 0 if the combination of components $\alpha \cap \mathbf{c}_1$ forms no products of complexity $c$ and 1 if $\alpha \cap \mathbf{c}_1$ forms one product of complexity $c$. (Occasionally, the same combination of components $\alpha \cap \mathbf{c}_1$ forms multiple products: for example, in our gastronomy data, beef, butter and onion together form two distinct recipes of length three. In such cases, $\mathrm{prod}(\alpha \cap \mathbf{c}_1)$ takes the value 2 if $\alpha \cap \mathbf{c}_1$ forms two products, and so on.) The mean usefulness of component $\alpha$, $\bar{u}_\alpha(n, c)$, is the average of $u_\alpha(\mathbf{n}, c)$ over all subsets $\mathbf{n}_1 \subseteq \mathbf{N}_1$; there are $\binom{N-1}{n-1}$

such subsets. Therefore,

$$\overline{u}_\alpha(n,c) = 1/\binom{N-1}{n-1} \sum_{\boldsymbol{n}_1 \subseteq N_1} u_\alpha(\boldsymbol{n},c), \tag{4}$$

$$= 1/\binom{N-1}{n-1} \sum_{\boldsymbol{n}_1 \subseteq N_1} \sum_{\boldsymbol{c}_1 \subseteq \boldsymbol{n}_1} \mathrm{prod}(\alpha \cap \boldsymbol{c}_1) - \mathrm{prod}(\boldsymbol{c}_1). \tag{5}$$

Consider some particular combination of components $\boldsymbol{c}_1{}'$. The double sum above will count $\boldsymbol{c}_1{}'$ once if $c = n$, but multiple times if $c < n$, because $\boldsymbol{c}_1{}'$ will belong to multiple sets $\boldsymbol{n}_1$. How many? In any set $\boldsymbol{n}_1$ that contains $\boldsymbol{c}_1$, there are $n - c$ free elements to choose, from $N - c$ other components. Therefore the double sum will count every combination $\boldsymbol{c}_1$ a total of $\binom{N-c}{n-c}$ times, and

$$\overline{u}_\alpha(n,s) = \binom{N-c}{n-c}/\binom{N-1}{n-1} \sum_{\boldsymbol{c}_1 \subseteq N_1} \mathrm{prod}(\alpha \cap \boldsymbol{c}_1) - \mathrm{prod}(\boldsymbol{c}_1) \tag{6}$$

$$= N/n \binom{n}{c}/\binom{N}{c} \overline{u}_\alpha(N,c). \tag{7}$$

The same must be true when we replace $n$ by $n'$, and therefore

$$\overline{u}_\alpha(n,c) \, n/\binom{n}{c} = \overline{u}_\alpha(n',c) n'/\binom{n'}{c}. \tag{8}$$

The usefulness $u_\alpha(\boldsymbol{n}, c)$ of some specific choice of components $\boldsymbol{n}$ is an unbiased estimate of the mean usefulness $\overline{u}_\alpha(n,c)$ averaged over all possible sets $\boldsymbol{n}$. This estimate is equivalent to taking a sample size of one in the average in Eq. (4). This can be a good estimate for two reasons: the samples are highly correlated, and the number of possible samples approaches one as $n$ approaches $N$. Therefore the equation for mean usefulness can give an estimation for usefulness,

$$u_\alpha(\boldsymbol{n},c) \, n/\binom{n}{c} \simeq u_\alpha(\boldsymbol{n}',c) n'/\binom{n'}{c}. \tag{9}$$

When the number of components is big compared to the product size ($n, n' \gg c$), we can approximate $\binom{n}{c}$ and $\binom{n'}{c}$ by $n^c$ and $n'^c$, and

$$u_\alpha(\boldsymbol{n},c) / n^{c-1} \simeq u_\alpha(\boldsymbol{n}',c)/ n'^{c-1}. \tag{10}$$

Equation (8) is exact even in the presence of correlations between the occurrence of different components in products; at no point in our proof did we assume component independence. Therefore correlations do not impact the accuracy of our forecasted mean, though they can lead to more fluctuations around the mean, or less precision. Typical fluctuations can be seen for two gastronomy ingredients in Fig. 1b.

**Forecasting crossovers in usefulness**. Here we show how we can forecast the usefulness of components at stage $n'$ from information we have at some earlier stage $n$, where $n$ is the number of components we have acquired. As in Fig. 3, we have a set $\boldsymbol{k}$ of $k = 127$ ingredients in a small kitchen—almond to fenugreek—and a set $\boldsymbol{K}$ of $K = 381$ ingredients in a big kitchen—almond to zucchini.

In the small kitchen, we can make a total of 597 recipes. Of these 597 recipes, 43 contain cayenne, but they are not all equally complex. Two of the 43 recipes contain one ingredient (namely, cayenne itself) and have complexity one; one recipe contains two ingredients and has complexity two; 18 contain three ingredients and have complexity three; and so on. Similarly, 89 of the 597 recipes contain cocoa: six have complexity one; 22 have complexity two; and so on. Substituting these values into Eq. (2), we can estimate the mean usefulness of these two components at different stages as

$$\overline{u}_{\mathrm{ca}}(n'|\boldsymbol{k}) \simeq 2 + x + 18x^2 + 12x^3 + 8x^4 + x^5 + x^7$$

and

$$\overline{u}_{\mathrm{co}}(n'|\boldsymbol{k}) \simeq 6 + 22x + 37x^2 + 16x^3 + 8x^4, \tag{11}$$

where $x = n'/127$. As expected,

$$\overline{u}_{\mathrm{ca}}(k|\boldsymbol{k}) = 43$$

and

$$\overline{u}_{\mathrm{co}}(k|\boldsymbol{k}) = 89. \tag{12}$$

In the big kitchen, we can make a total of 56,498 recipes. Of these, 7950 contain

cayenne and 4801 contain cocoa. Again using Eq. (2),

$$\overline{u}_{\mathrm{ca}}(n'|\boldsymbol{K}) \simeq 2 + 19x + 64x^2 + \cdots + 2x^{28} + 2x^{30}$$

and

$$\overline{u}_{\mathrm{co}}(n'|\boldsymbol{K}) \simeq 6 + 54x + 195x^2 + \cdots + 2x^{20} + 3x^{21}, \tag{13}$$

where $x = n'/381$. As expected,

$$\overline{u}_{\mathrm{ca}}(K|\boldsymbol{K}) = 7950$$

and

$$\overline{u}_{\mathrm{co}}(K|\boldsymbol{K}) = 4801. \tag{14}$$

So far, none of this is surprising. The punchline is that we can estimate the usefulness of components in the big kitchen from what we know about our small kitchen. To do so, we simply evaluate the small-kitchen polynomials at the big-kitchen stage:

$$\overline{u}_{\mathrm{ca}}(K|\boldsymbol{K}) \simeq \overline{u}_{ca}(K|\boldsymbol{k}) \simeq 3569$$

and

$$\overline{u}_{\mathrm{co}}(K|\boldsymbol{K}) \simeq \overline{u}_{\mathrm{co}}(K|\boldsymbol{k}) \simeq 1485. \tag{15}$$

This predicts the crossover of cayenne and cocoa in Fig. 3. In log terms, these estimates are accurate to be within 9 and 13% of the true values. The reason we consider log usefulness is because the size of the product space grows combinatorially with the number of acquired components, as can be seen in Fig. 1.

**Data availability**. All relevant data are available in ref. [23] or from the corresponding authors upon reasonable request.

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

## Acknowledgements

We acknowledge support from the European Commission FP7 grant 611272 "Growth and innovation policy modelling (GROWTHCOM)" and the Porter Foundation.

## Author contributions

The methodology was planned by T.F., R.F. and M.R. Analytics were done by T.F. and R.F. Data collection was done by R.P. and T.F. Data analysis was done by T.F. and R.P. Figure images were made by T.F. The manuscript was written by T.F., M.R. and R.F. Implications of the results were conceived by M.R. and T.F.

## Additional information

**Competing interests:** The authors declare no competing financial interests.

