## [Peer Review File · Nature Communications]

Reviewers' comments:

Reviewer #1 (Remarks to the Author):

Referee report on the paper "Serendipity and strategy in rapid innovation". In this paper authors represent innovation as an lego model of building blocks from which new innovations are developed. They approach the problem through idea that not all the "blocks" have similar value in different stages of innovation process. They develop a model based on statistics of available information of how different blocks are used, and present how one can use this model to evaluate value of building blocks in the future. They test their model on three different data sets that are: words and letters, food recipes and ingredients and software products and development tools. The paper is clearly written, model is elegant and simple (some may argue too simple). Results are also clear and methodology makes sense. One could wish that authors had some additional dataset, because Words are not that interesting due to very small number of crossovers, while software products give great results, but the sample is relatively small. My assessment is that this paper is worthy of publishing in Nature communications after some responses from the authors.

1. Lego building block of innovation in itself is not a brand-new idea. Cesar Hidalgo and Ricardo Hausmann had a series of papers which were building on similar idea in macro economical context. They were followed by Pietronero group which built on this idea. There is a clear difference between this paper which is taking "micro economical" approach as opposed to these papers. Moreover this paper is focused on innovation and not economic complexity, and this papers put in perspective the validity of authors approach. Moreover even if not motivated by this papers - appropriate links should be made (referee is not an author of any of listed papers). The list of papers that SHOULD be mentioned is bellow.

- a. CA Hidalgo, B Klinger, AL Barabási, R Hausmann Science 317 (5837), 482 (2007)
- b. CA Hidalgo, R Hausmann Proceedings of the National Academy of Sciences 106 (26), 10570-10575 (2009)
- c. Tacchella, Andrea and Cristelli, Matthieu and Caldarelli, Guido and Gabrielli, Andrea and Pietronero, Luciano Scientific reports 2, 723 (2012)
- d. Cristelli, Matthieu and Gabrielli, Andrea and Tacchella, Andrea and Caldarelli, Guido and Pietronero, Luciano, PLoS one 8 (8), e70726 (2013)

2. Although the paper is clearly written it is not clear that the statistics is taken at time t and not from the whole history (including future) that should be stressed more clearly.

3. I would like authors to comment a bit on the problem of correlations, since clearly some blocks are going to be used more together than some other. Could they create artificial data with correlations built in so that they would see if their model still gives reasonable results?

Reviewer #2 (Remarks to the Author):

This paper discusses the process of innovation as one in which an entity is collecting a number of components that can be recombined in order to make products. The main results I summarize as follows:

- Identifying differences in crossovers in average component usefulness in the datasets considered, i.e. the 'bumps charts' in Fig. 2
- The formulation of strategies (ranging from short-sighted to far-sighted) in order to maximize the number of products one can make on average given a number a components.
- The proof of an invariant (Section 3b), showing how the average usefulness of a component in products with given complexity s grows with the number of components owned (eq. 1)

The question the paper and its key results are very interesting, useful and novel. The idea that serendipity may be explained by a former collection latent components (not or rarely used in the past) is appealing and new to the literature. I also value the use of three rather 'uncommon' data sets. I am therefore inclined to recommend publication.

Yet, I have some questions that have to be resolved in my view.

I start with my main critique and then list some further issues for improvement.

The conservation law proved in Section 3b provides insight in how the average usefulness of components for products with complexity s changes in time, and appears to be a truly novel and interesting result that may be the basis of future research.

The claim that this conservation law is also useful for deploying a certain strategy however is less convincing, and it is unclear how a forecast such as the one made in Section 3.c can be made with only information about existing components, as the authors seem to claim or imply in Section 2. Far-sighted strategy. To engage in a far-sighted strategy, one needs the 'full recipe book' in order to compute the average usefulness of components, because – as the authors do say – a far-sighted strategy is based on information on a component's usefulness and valence, which is "the average complexity of the products it appears in" (p. 3). This variable is an average over all possible products

it appears: “The valence v_α of component α is the average complexity of the products it appears in at stage N , when we have all N components.” (p. 3). Therefore, it seems a bit odd to compare a myopic and a far-sighted strategy in this way, because you assume that a far-sighted firm has global knowledge of the recipe book.

To resolve this weakness in the paper, I see three options:

- The authors make clear and defend that they make the heroic assumption that a firm somehow can know the valence of a component.

- The authors make clear how valence just defined (average complexity of the products a component appears in at stage N) of a component can be reliably estimated from the “local valence” which would be average complexity of the products a component appears in at stage n .

- The authors drop the part on strategy and focus only on the other results (the invariant and their new theory of serendipity). If so, obviously, the paper has to be rewritten without any reference to strategy.

I reckon that option 1 is rather unsatisfactory given the strong assumption about “global knowledge” it has to make, which goes against the evolutionary nature of the theory itself. That is why I urge the authors to explore option 2 or 3. Although option 3 would remove one of the paper's contributions, I would still recommend publication if the authors decide to remove this part.

(Alternatively – option 4 – is that I may be complexity mistaken and if so, I invite the authors to explain why I am mistaken about this in the response letter).

Other comments:

1. The paper is poorly embedded in the literature. First and foremost, the paper understands technology evolution as recombinant with components being added one by one. This idea underlies some economic growth models, in particular, Tacchella et al. (2012), Hidalgo & Hausmann (2009) and Hausmann & Hidalgo (2011). Engaging with this literature would certainly increase its relevance and create a wider readership. Furthermore, how does your model relate to the earlier NK-model by Kauffman (1993) and especially the generalized NK-model by Altenberg (1994)?

2. Throughout the paper the terms usefulness and average usefulness are used interchangeably, leading to confusion.
3. The terms isochronic and anisochronic in Section 1. Bumps Charts are used only once in the text and do not seem to have a clear further function.
4. The claims that more valent components rise in the bump charts and vice versa in Section 1. Valence are not quantified.
5. Notation in Section 3.c: the 'conditioning on a set notation', i.e. $|k$ and $|K$ suggests that one takes set N to be k or K . In other words, a subset of all ingredients is considered (as described in the text). But this seems to imply that the averages of the usefulness are also taken over the subset k only, meaning considering the average usefulness of α over all subsets of size $(n-1)$ of set k and K respectively. But equation 3 holds only in cases where average are taken over a similar set (namely the entire set N in Section 3b). I may be mistaken in my reading here. Please clarify.
6. Why is log-usefulness the natural unit of measure, and why is 9-11% accurate? It seems that in order to make claims about the accuracy of such forecasts one would have to look at averages/distributions for all possible k and K (since the prediction is about averages) .
7. The short-sighted strategy seems to be choosing the component with highest average usefulness at a given n (equivalent to choosing a component that intersect the diagonals in Figure 2). An effective greedy strategy however would be to take the component with maximal usefulness for the set currently owned, i.e. not the average usefulness.
8. I value a lot the use of the three datasets. This makes the paper appealing and much more convincing that having used just one dataset. Nevertheless, from an empirical point of view, it would be interesting to discuss the differences between the data as well. What, for example, does the increased number of crossovers tell us about the technology system and its properties compared to language or gastronomy, which appear different in this respect? Does this has to do with the modular nature of technology (Arthur 2009).

References

Altenberg, L. (1994) Evolving better representations through selective genome growth, in: Schaffer, J.D., Schwefel, H.P., Kitano H. (eds.) Proceedings of the IEEE World Congress on Computational Intelligence. Piscataway, NJ: IEEE, pp. 182–187.

Arthur, WB (2009) The Nature of Technology: What it Is and How it Evolves, New York: Free Press

Hausmann, R., Hidalgo, C. (2011) The network structure of economic growth, Journal of Economic Growth 16, pp. 309-342.

Hidalgo, C., Hausmann, R. (2009) The buildings blocks of economic complexity, PNAS 106(26), pp. 10570-10575.

Kauffman, S.A. (1993) The Origins of Order. New York & Oxford: Oxford University Press.

Tacchella Andrea, Matthieu Cristelli Guido Caldarelli Andrea Gabrielli Luciano Pietronero (2012) A New Metrics for Countries' Fitness and Products' Complexity, Scientific Reports 2, Article number: 723

Response to *Nature Communications* referee reports

In what follows, the referees' comments are in black and our responses and revisions are in blue.

We found the referees' reports particularly helpful in this instance to making our manuscript a better and clearer research paper. We believe we have satisfactorily addressed all of their concerns.

Reviewer #1

In this paper authors represent innovation as an lego model of building blocks from which new innovations are developed. They approach the problem through idea that not all the "blocks" have similar value in different stages of innovation process. They develop a model based on statistics of available information of how different blocks are used, and present how one can use this model to evaluate value of building blocks in the future. They test their model on three different data sets that are: words and letters, food recipes and ingredients and software products and development tools.

The paper is clearly written, model is elegant and simple (some may argue too simple). Results are also clear and methodology makes sense. One could wish that authors had some additional dataset, because Words are not that interesting due to very small number of crossovers, while software products give great results, but the sample is relatively small.

My assessment is that this paper is worthy of publishing in *Nature Communications* after some responses from the authors.

Referee's comment	Our response to the comment	Revisions we made to the manuscript
1. Lego building block of innovation in itself is not a brand-new idea. Cesar Hidalgo and Ricardo Hausmann had a series of papers which were building on similar idea in macro economical context. They were followed by Pietronero group which built on this idea. There is a clear difference between this paper which is taking "micro economical" approach as opposed to these papers. Moreover this paper is focused on innovation and not economic complexity, and this papers put in perspective the validity of authors approach. Moreover even if not motivated by this papers - appropriate links should be made (referee is not an author of any of listed papers). The list of papers that SHOULD be mentioned is below. C. Hidalgo et al., Science 317 (5837), 482 (2007). C. Hidalgo, R Hausmann PNAS 106, 10570 (2009). M. Cristelli et al., PloS one 8, e70726 (2013). A. Tacchella et al., Scientific reports 2, 723 (2012).	We are grateful to the reviewer for bringing these papers to our attention. We already do cite one of his four suggested references (Tacchella 2012). Regarding the others, we agree—they help better embed our work into related research. We now cite all four, and a couple of others (shown right), and incorporate them into our manuscript.	We added the following five references, and cite them in the first paragraph after the abstract.  • C. Hidalgo, B. Klinger, A.-L. Barabasi, R. Hausmann, 'The product space conditions the development of nations', Science 317, 482 (2007). • C. Hidalgo, R. Hausmann, 'The buildings blocks of economic complexity', Proc Natl Acad Sci 106, 10570 (2009). • R. Hausmann, C. Hidalgo, 'The network structure of economic output', J Econ Growth 16, 309 (2011). • M. Christelli et al., 'Measuring the intangibles: a metrics for the economic complexity of countries and products', PLoS ONE 8, e70726 (2013). • W. Arthur, The Nature of Technology: What it Is and How it Evolves (Allen Lane, 2009). We have added and modified the following sentences to the first paragraph after the abstract: "Research on macro-economic development suggests that more complex, diverse or re-purposable production capabilities result in greater economic growth at a national level. At a micro-economic level, there is a perennial tension between a strategic approach..."
2. Although the paper is clearly written it is not clear that the statistics is taken at time t and not from the whole history (including future) that should be stressed more clearly.	We agree, and we now stress this point more clearly.	At the end of the paragraph "Why crossovers happen", we added the text and equation: "Because the usefulness is an unbiased estimate of its mean, <new displayed equation>, entirely from information we have at the present."

3. I would like authors to comment a bit on the problem of correlations, since clearly some blocks are going to be used more together than some other. Could they create artificial data with correlations built in so that they would see if their model still gives reasonable results?	The reviewer is right to suggest that correlations do occur. Fortunately, the conservation law that we prove in §3.b is strictly true even in the presence of correlations.	We add the following paragraph to the bottom of §3.b: “Correlations. Eq. (4) is exact even in the presence of correlations between the occurrence of different components in products; at no point in our proof did we assume component independence. Therefore correlations do not impact the accuracy of our forecasted mean, though they can lead to more fluctuations around the mean, or less precision.”
--	--	---

Reviewer #2

This paper discusses the process of innovation as one in which an entity is collecting a number of components that can be recombined in order to make products.

The main results I summarize as follows:

Identifying differences in crossovers in average component usefulness in the datasets considered, i.e. the ‘bumps charts’ in Fig. 2.

The formulation of strategies (ranging from short-sighted to far-sighted) in order to maximize the number of products one can make on average given a number a components.

The proof of an invariant (Section 3b), showing how the average usefulness of a component in products with given complexity s grows with the number of components owned (eq. 1).

The question the paper and its key results are very interesting, useful and novel. The idea that serendipity may be explained by a former collection latent components (not or rarely used in the past) is appealing and new to the literature. I also value the use of three rather ‘uncommon’ data sets.

I am therefore inclined to recommend publication.

Yet, I have some questions that have to be resolved in my view. I start with my main critique and then list some further issues for improvement.

Main critique

Referee’s comment

The conservation law proved in Section 3b provides insight in how the average usefulness of components for products with complexity s changes in time, and appears to be a truly novel and interesting result that may be the basis of future research.

The claim that this conservation law is also useful for deploying a certain strategy however is less convincing, and it is unclear how a forecast such as the one made in Section 3.c can be made with only information about existing components, as the authors seem to claim or imply in Section 2. Far-sighted strategy. To engage in a far-sighted strategy, one needs the ‘full recipe book’ in order to compute the average usefulness of components, because – as the authors do say – a far-sighted strategy is based on information on a component’s usefulness and valence, which is “the average complexity of the products it appears in” (p. 3). This variable is an average over all possible products it appears: “The valence v_α of component α is the average complexity of the products it appears in at stage N , when we have all N components.” (p. 3). Therefore, it seems a bit odd to compare a myopic and a far-sighted strategy in this way, because you assume that a far-sighted firm has global knowledge of the recipe book.

To resolve this weakness in the paper, I see three options:

- Option 1. The authors make clear and defend that they make the heroic assumption that a firm somehow can know the valence of a component.
- Option 2. The authors make clear how valence just defined (average complexity of the products a component appears in at stage N) of a component can be reliably estimated from the “local valence” which would be average complexity of the products a component appears in at stage n .
- Option 3. The authors drop the part on strategy and focus only on the other results (the invariant and their new theory of serendipity). If so, obviously, the paper has to be rewritten without any reference to strategy.

I reckon that option 1 is rather unsatisfactory given the strong assumption about “global knowledge” it has to make, which goes against the evolutionary nature of the theory itself. That is why I urge the authors to explore option 2 or 3. Although option 3 would remove one of the paper's contributions, I would still recommend publication if the authors decide to remove this part.

- (Alternatively – option 4 – is that I may be completely mistaken and if so, I invite the authors to explain why I am mistaken about this in the response letter).

Our response to the comment

Overall this referee has given an *excellent* critique of our work. His comments are insightful and penetrating, and thinking about and responding to them has helped us make this paper clearer and more impactful. Few referees have given others papers of our as much consideration.

The referee is right to point out the possibility for confusion on whether we assume knowledge of the future. We do not, and we have used this opportunity to eliminate potential confusion for other readers.

Options 2 and 4 above are the correct options: we do not assume “omniscience”. In other words, we do not assume that a firm can somehow know the set of future products that it can make with yet-to-be-acquired components.

Confusion stems from two things: (i) our simplified presentation of valence; and (ii) a need for more care in the distinction between usefulness and mean usefulness.

Regarding valence: like usefulness, valence can be calculated at stages $n < N$; it can only be known with *certainty* at stage N . We did not show this time dependence for the sake of simplicity. Since valence is just the average size of the product containing α , if we can estimate the number of products of a given size at stage n (by eq. (1)), we can estimate valence. We now make this more clear in our definition and discussion of valence.

Regarding knowledge of the “full recipe book”: even though we posit the existence of N (the number of components in “God’s own cupboard”), and use it our proofs, we do not actually need to know N at all, nor the products makeable from yet-to-be-discovered components. We resolve this by recasting the old eqs., which were in terms of mean usefulness (\bar{u}), in terms of usefulness (u).

Changes we made to the manuscript

(i) We clarified the definition of valence in the middle of the “Valence” paragraph:

“The valence $v \alpha (n)$ of component α is the average complexity of the products it appears in at stage n . Think of the valence as the typical number of co-stars a component performs with, plus one. We show the usefulness and valence at stage N for different components in Fig. 5ABC and Figs. 6 and 7 top.”

(ii) We added the following paragraph towards the end of §Methods 3.b:

Usefulness. The usefulness $u \alpha (n,s)$ of some specific choice of components n is an unbiased estimate of the mean usefulness $\bar{u} \alpha (n, s)$ averaged over all possible sets n . This estimate is equivalent to taking a sample size of one in the average in (3). This can be a good estimate for two reasons: the samples are highly correlated, and the number of possible samples approaches one as n approaches N . Therefore the equation for mean usefulness gives an estimation for usefulness,

<equation here>.
We then bring this down into our Results: At the end of the paragraph “Why crossovers happen”, we added the text and equation:

“Because the usefulness is an unbiased estimate of its mean,
<new displayed equation>,
entirely from information we have at the present. ”

In addressing the issues below, we also touch upon these points further.

Further issues for improvement

Referee’s “other comments”	Our response to the comment	Revisions we made to the manuscript
1. The paper is poorly embedded in the literature. First and foremost, the paper understands technology evolution as recombinant with components being added one by one.	We are grateful to the reviewer for bringing some of these papers to our attention. We already do cite one of his suggested references (Tachella 2012). Regarding the others, we agree	We added the following five references, and cite them in the first paragraph after the abstract.  • C. Hidalgo, B. Klinger, A.-L. Barabasi, R. Hausmann, ‘The product space

This idea underlies some economic growth models, in particular, Tacchella et al. (2012), Hidalgo & Hausmann (2009) and Hausmann & Hidalgo (2011). Engaging with this literature would certainly increase its relevance and create a wider readership. Furthermore, how does your model relate to the earlier NK-model by Kauffman (1993) and especially the generalized NK-model by Altenberg (1994)?	that several help embed our work into related research. We now cite four of his suggestions, and a couple of others (shown right), and incorporate them into our manuscript. We appreciate the link to the work on evolving the search space while running a genetic algorithm; especially Altenberg (1994). The analogy between high-fitness phenotypes and successful innovations is clear, and has been often remarked upon; but we believe the referee is suggesting a different analogy between additional genes added to a genome and an enhanced capability for innovation (so that extra loci are serving the function of extra ingredients in the “cupboard” in our work). We believe that our measure of the utility of a cupboard of ingredients, in terms of how numerous the resulting recipes are, is both simpler and (we believe) more direct than the Altenberg work, which shows that strategically expanding the genome can make a system more evolvable towards high fitness states. We hope to explore those intriguing connections between different fields in future work, but we feel that at this stage, trying to make those (to us, as yet unclear) links would make our presentation more opaque.	conditions the development of nations', Science, 317, 482 (2007).  ● C. Hidalgo, R. Hausmann, 'The buildings blocks of economic complexity,' Proc Natl Acad Sci, 106, 10570 (2009). ● R. Hausmann, C. Hidalgo, 'The network structure of economic output', J Econ Growth, 16, 309 (2011). ● M. Christelli et al., 'Measuring the intangibles: a metrics for the economic complexity of countries and products', PLoS ONE, 8, e70726 (2013). ● W. Arthur, The Nature of Technology: What it Is and How it Evolves (Allen Lane, 2009). We have added and modified the following sentences to the first paragraph after the abstract: “Research on macro-economic development suggests that more complex, diverse or re-purposable production capabilities result in greater economic growth at a national level. At a micro-economic level, there is a perennial tension between a strategic approach...”
2. Throughout the paper the terms usefulness and average usefulness are used interchangeably, leading to confusion.	The reviewer rightly points out that this distinction needs to be emphasized. We have updated the wording in the definition of usefulness in the “Usefulness” paragraph, clarified both terms in the “Usefulness experiment” paragraph, and have at various places been more careful in our choice of “usefulness” versus “mean usefulness”. The potential for confusion stemmed a lot we imagine from the old eqs. (1) and (2) being written in terms of mean usefulness, which we now have cast in terms of usefulness.	We inserted the underlined words in the definition of usefulness in the “Usefulness” paragraph: “usefulness of α given possession of α and a specific set of $n - 1$ other components...”. At the end of the “Usefulness experiment” paragraph, we added the following text: “As Fig. 1 shows for gastronomy, this is true for both the mean usefulness and the usefulness itself, measured for a specific random ordering of components (points). What the mean usefulness does, the usefulness tends to do also, because it is an unbiased estimate of its mean.”. We also cast the old eqs. (1) and (2), which were in terms of mean usefulness, into the new eqs. (1) and (2), which are in terms of usefulness.
3. The terms isochronic and anisochronic in Section 1. Bumps Charts are used only once in the text and do not seem to have a clear further function.	We agree, we introduced the terms but don't make later use of them.	We have removed this sentence, namely, the last sentence in the “Bumps charts” paragraph.

4. The claims that more valent components rise in the bump charts and vice versa in Section 1. Valence are not quantified.	To quantify this, we added a new plot, described to the right.	We added the following plot as a new Figure (what is now Fig. 4), with the caption: "FIG. 4: Valence. Here we show the middle panel of Fig. 2, but with the component curve colors set by component valence at stage N. More valent components tend to rise in relative usefulness, and less valent components tend to fall." 5. Notation in §3.c: the 'conditioning on a set notation', i.e. k and K suggests that one takes set N to be k or K. In other words, a subset of all ingredients is considered (as described in the text). But this seems to imply that the averages of the usefulness are also taken over the subset k only, meaning considering the average usefulness of α over all subsets of size $(n-1)$ of set k and K respectively. But equation 3 holds only in cases where average are taken over a similar set (namely the entire set N in §3b). I may be mistaken in my reading here. Please clarify.	The better distinction between usefulness and mean usefulness should make this more transparent, as shown right.	We added the following paragraph towards the end of §3.b: Usefulness. The usefulness $u_\alpha(n,s)$ of some specific choice of components n is an unbiased estimate of the mean usefulness $\bar{u}_\alpha(n,s)$ averaged over all possible sets n. This estimate is equivalent to taking a sample size of one in the average in (3). This can be a good estimate for two reasons: the samples are highly correlated, and the number of possible samples approaches one as n approaches N. Therefore the equation for mean usefulness gives an estimation for usefulness, <equation here> This should clarify the examples in the next subsection, §3.c, since there is no implicit reference to knowing N and not just n (K and not just k) due to taking averages of k from K.
6. Why is log-usefulness the natural unit of measure, and why is 9-11% accurate? It seems that in order to make claims about the accuracy of such forecasts one would have to look at averages/distributions for all possible k and K (since the prediction is about averages).	We agree with the referee that our use of log usefulness needs a bit of explanation, which we have added (see right). The reason that our accuracies are sufficient is that we simply want to predict crossovers. In the small kitchen, cayenne is 48% as useful as cocoa. In the big kitchen, cayenne is 166% as useful as cocoa. Using just the information available in the smaller kitchen, we correctly predict that cayenne goes from less useful to more useful.	We removed the "natural unit of measure" clause and added the sentence "We consider log usefulness because the size of the product space grows combinatorially with the number of acquired components, as can be seen in Fig. 1." We also corrected a typo: 11% should have been 13%.
7. The short-sighted strategy seems to be choosing the component with highest average usefulness at a given n (equivalent to choosing a component that intersect the diagonals in Figure 2). An effective greedy strategy however would be to take the component with maximal usefulness	The referee's second sentence is correct. The scope for mistakenly concluding the first sentence should be addressed by casting the old eqs. (1) and (2), which were in terms of mean usefulness, into the new eqs. (1) and (2), which are in terms of usefulness.	See our response to point 2 above.

for the set currently owned, i.e. not the average usefulness.		
8. I value a lot the use of the three datasets. This makes the paper appealing and much more convincing that having used just one dataset. Nevertheless, from an empirical point of view, it would be interesting to discuss the differences between the data as well. What, for example, does the increased number of crossovers tell us about the technology system and its properties compared to language or gastronomy, which appear different in this respect? Does this has to do with the modular nature of technology (Arthur 2009).	What causes some datasets to have more crossovers than others is an interesting question. While we don't have a complete answer, we expect that a dataset with a broad spectrum of valences (that is to say, a dataset in which some components preferentially belong to complex products, and others to simple products) is likely to have many crossovers.	We added the following text to the end of the "Strategy comparison" paragraph in §2: "Why do some datasets have more crossovers than others? Our results suggest that a dataset with a broader spectrum of valences tends to have more crossovers. For language, gastronomy and technology the standard deviations of valence $v \propto (N)$ are 0.32, 2.0 and 18."

REVIEWERS' COMMENTS:

Reviewer #1 (Remarks to the Author):

I believe that the manuscript "Serendipity and strategy in rapid innovation" is now ready to be published. Authors took all of the criticism seriously and responded to it satisfactorily. I am happy to recommend the paper for the publication in Nature Communications.

Reviewer #2 (Remarks to the Author):

Dear authors,

The revisions have improved the paper. However, some confusion remains.

In response to point 7:

If indeed the greedy strategy is about usefulness and not average usefulness, the interpretation of it being equivalent to acquiring the components that intersect the diagonal in Fig 2 is wrong., since Fig 2 depicts the average usefulness. In addition, do the plots in Fig 5 show the average over all greedy strategies with a particular starting component, 1 particular instance of a greedy strategy (for 1 single starting component), or the strategy that chooses the component with maximal mean usefulness at a given time (which is then not the proposed strategy).

In response to point 5:

The text suggests estimates are being made of the usefulness, but the equations show the average usefulness (i.e. \bar{u}).

To my understanding, in notation for usefulness the n in $u(n)$ denotes a set,

and in notation for average usefulness, the n in $\bar{u}(n)$ denotes a natural number.

This leads to trouble in understanding the very last equation. If I understood correct the the paper, the reasoning is that

$u_{ca}(k)$ is an estimate of $\bar{u}_{ca}(127)$ so that

$u_{ca}(n'|k)|_{x=3} \approx 3569$ is an estimate of $\bar{u}_{ca}(381)$

which is then an estimate of $\bar{u}(K)$

This would also imply that the forecast holds for any set K' (containing k ?) with size 381. Can one show the distribution of the values of all sets K' , and then show the forecast is reasonable?

Again, I might be mistaken in my understanding here.

More in general:

The claim that equation 1 follows from equation 4 is in my view an assumption unless shown (possibly numerically), for example in a way similar to suggested above; how are the usefulness values for particular sets distributed around the mean? Figure 1 only shows 2 samples, and one expects big variations for systems with high correlations (as noted in the paper). Another option is to justify such an assumption in the results section, where again only 1 sample is shown. In any case, the authors should be more explicit in the assumptions they make.

Response to *Nature Communications* 2nd referee reports

In what follows, the referee's comments are in black and our responses and revisions are in blue.

We believe we have satisfactorily addressed all of their concerns.

Reviewer #2

Referee's comment	Our response to the comment
Dear authors, The revisions have improved the paper. However, some confusion remains. In response to point 7: If indeed the greedy strategy is about usefulness and not average usefulness, the interpretation of it being equivalent to acquiring the components that intersect the diagonal in Fig 2 is wrong., since Fig 2 depicts the average usefulness. In addition, do the plots in Fig 5 show the average over all greedy strategies with a particular starting component, 1 particular instance of a greedy strategy (for 1 single starting component), or the strategy that chooses the component with maximal mean usefulness at a given time (which is then not the proposed strategy).	We have clarified the explanation on usefulness and average usefulness in the section titled "conservation law", and also given a fuller explanation of the far-sighted strategy at this point. The referee is correct that in Fig 2 we are plotting average usefulness, so the strategy we describe is only an attempt to acquire the components on the diagonal (or vertical) of Figure 2. We have amended the description to of the strategy, in the discussion section (paragraphs 2 and 3), to capture this. Figure 5 shows particular instances of the innovation history, and therefore illustrate the (now, hopefully, more clearly described) far- and long-sighted strategies in the text.
In response to point 5: The text suggests estimates are being made of the usefulness, but the equations show the average usefulness (i.e. \bar{u}). To my understanding, in notation for usefulness the n in $u(n)$ denotes a set, and in notation for average usefulness, the n in $\bar{u}(n)$ denotes a natural number.	The referee is correct, and in the previous explanation, we did not draw the distinction between u and \bar{u} clearly enough in our explanations. The conservation law (eq. (1)) and the projection equation (eq. (2)) indeed concern the mean usefulness. We now describe exactly how to make use of these equations, employing only data available at the present time (see section "Conservation law").
This leads to trouble in understanding the very last equation. If I understood correct the paper, the reasoning is that $u_{\{ca\}}(k)$ is an estimate of $\bar{u}_{\{ca\}}$ (127) so that $u_{\{ca\}}(n' k) _{x=3} \approx 3569$ is an estimate of $\bar{u}_{\{ca\}}$ (381) which is then an estimate of $\bar{u}(K)$ This would also imply that the forecast holds for any set K' (containing k?) with size 381. Can one show the distribution of the values of all sets K', and then show the forecast is reasonable?	Again, the referee is correct (including the point that the forecast can be done for any K' a superset of k). The more detailed explanation in section "Conservation law" covers these points. The referee is correct that using a particular history to estimate the mean over all histories (and vice versa) introduces noise into the predictions. We believe that the particular examples we show (the comparison at the end of the methods section, and the particular histories superposed on the mean histories in lower middle panel of Figure 1) illustrate the magnitude of the noise introduced by these approximations.
More in general: The claim that equation 1 follows from equation 4 is in my view an assumption unless shown (possibly numerically), for example in a way similar to suggested above; how are the usefulness values for particular sets distributed around the mean? Figure 1 only shows 2 samples, and one expects big variations for systems with high correlations (as noted in the paper). Another option is to justify such an assumption in the results section, where again only 1 sample is shown. In any case, the authors should be more explicit in the assumptions they make.	We agree that the relation between these two equations was not clearly laid out in the previous version of the manuscript. Again, we hope that the more detailed explanation added in section "Conservation law" addresses this point. We have not yet performed a broad systematic study of the noise (or uncertainty) introduced by using particular histories to estimate the mean (which will form the subject of a follow up paper), but have illustrated it with the cases mentioned above.